# Finite blocklength feedback approach for multi-antenna wireless channels

**Qiang Huang**, **Tao Song** *

Geely University of China, Chengdu, Sichuan, China

These authors contributed equally to this work.
* songtao@guc.edu.cn

**Data Availability Statement:** Our work focuses on theoretical analysis, no experimental datasets are used, and the images in the paper are based on some numerical results only.

**Funding:** the authors received no specific funding for this work.

## Abstract

Ultra-reliable low-latency communication (URLLC) is a key technology in future wireless communications, and finite blocklength (FBL) coding is the core of the URLLC. In this paper, FBL coding schemes for the wireless multi-antenna channels are proposed, which are based on the classical Schalkwijk-Kailath scheme for the point-to-point additive white Gaussian noise channel with noiseless feedback. Simulation examples show that the proposed feedback-based schemes almost approach the corresponding channel capacities.

## 1 Introduction

The rapid development of wireless communication technologies has driven the emergence of numerous innovative applications and services that demand ultra-reliable and low-latency communications (URLLC). To address the evolving requirements of industries such as industrial automation, healthcare, transportation, and virtual reality, the concept of URLLC has gained significant attention. URLLC, a fundamental aspect of the fifth-generation (5G) and beyond communication systems, aims to provide highly reliable and low-latency connectivity for mission-critical and latency-sensitive applications [1].

In practical wireless communication systems, finite blocklength (FBL) coding is a favorable way [2–4] for reducing the end-to-end communication latency, and the analysis of FBL coding receives a great deal of attention. To be specific, the second-order asymptotics for discrete memoryless case was first studied by Strassen [5]. Then Hayashi [6, 7] extended the result in [5] to more general cases by using the information spectrum method [8], and obtained the optimum second-order coding rate [6]. In addition, Tan [9] presented a unified treatment for asymptotic estimates in information theory with non-vanishing error probabilities, Zhou and Motani [10] provided a comprehensive review of recent advances in second-order asymptotics for lossy source coding. Recently, the bounds on the maximal transmission rate in FBL regime were given by Polyanskiy, Poor and Verdú in [11]. Subsequently, Truong, Fong and Tan [12] stated that channel with feedback link is an useful tool to construct the practical FBL coding scheme, namely the classical Schalkwijk-Kailath (SK) scheme [13]. The basic intuition of the classical SK scheme is that at each time instant, the receiver does minimum mean square estimation about the transmitted message, and sends his estimation back to the transmitter via a noiseless feedback channel. Then, the transmitter obtains the receiver's estimation error since

**Competing interests:** The authors have declared that no competing interests exist.

the he knows the real message. In the next time, the transmitter sends the receiver's estimation error in the last time to the receiver. It has been shown that the decoding error probability of the classical SK scheme double-exponentially decays as the codeword length increases, which indicates that to achieve a desired decoding error probability, the codeword length of the SK scheme is significantly short.

However, the classical SK coding scheme is mainly investigated in point-to-point additive white Gaussian noise (AWGN) channel, and the challenge to the application of SK scheme to practical wireless multi-antenna systems is as follows.

- The classical SK scheme [13] is designed for real-domain signals and noise. How to extend it to the complex-domain wireless static channel with single antenna equipped by the transceiver?

- How to further extend the above scheme to the same model with multiple antennas?

In this paper, we answer the above questions by extending the classical SK scheme to the single-input single-output (SISO)/single-input multiple-output (SIMO)/multiple-input single-output (MISO)/multiple-input multiple-output (MIMO) channels. The technical innovations are given below:

- For the SISO channel: By dividing the SISO channel with complex-valued channel parameters into two equivalent sub-channels with real-valued channel parameter, and by applying the classical SK scheme to each of the sub-channels, we obtain the SK-type FBL scheme for the SISO channel with noiseless feedback.

- For the SIMO channel: The SIMO channel can be transformed into SISO channel by using receiving beamforming technique. Then applying the SK-type FBL scheme for SISO channel to the obtained SISO channel, the SK-type FBL scheme for the SIMO channel is obtained.

- For the MISO channel: The MISO channel can be transformed into SISO channel by using precoding technique. Then applying the SK-type FBL scheme for SISO channel to the obtained SISO channel, the SK-type FBL scheme for the MISO channel is obtained.

- For the MIMO channel: The MIMO channel can be transformed into multiple parallel sub-channels by singular value decomposition technique. Then applying the SK-type FBL scheme for SISO channel to the each SISO sub-channel, the SK-type FBL scheme for the MIMO channel is obtained.

Organization: Formal definitions of the studied systems are given in Section 2. SK-type FBL coding schemes for these communication systems are shown in Section 3. Simulation examples are given in Section 4. Section 5 includes conclusions of this paper and discusses the future work.

*Assumptions*:

- We assume that all CSIs stay constant during the entire transmission.

- All CSIs are shared by the transceiver in the system.

*Notations*: Table 1 summarizes the symbols used in this paper.

## 2 Model formulation

The wireless static channels investigated in this paper are given in the following Fig 1, which consist of a transmitter and a receiver equipped with $T$ ($T \geq 1$) antenna(s) and $M$ ($M \geq 1$) antenna(s), respectively. In the following Fig 1, when ($T = M = 1$), ($T = 1$, $M \geq 2$), ($T \geq 2$,

**Table 1. Notations.**

| Notation | Meaning |
|---|---|
| $n$ and $\varepsilon$ | Codeword length and decoding error probability, respectively |
| $T$ and $M$ | Number of transmitting and receiving antennas, respectively |
| Boldface letter | A matrix or a vector |
| $\mathbf{I}_N$ | An $N \times N$ identity matrix |
| $\mathbf{Y}^n$ | Denotes $\mathbf{Y}_1, \mathbf{Y}_2, \ldots, \mathbf{Y}_n$ |
| $(\cdot)^{\mathcal{H}}$ | Conjugate transpose |
| $\det(\cdot)$ | Determinant of a square matrix |
| $\lvert \cdot \rvert$ | Modulus of a complex number or cardinality of a set |
| $\lVert \cdot \rVert$ | $l_2$-norm of a vector |
| $E(\cdot)$ and $\mathrm{Var}(\cdot)$ | Statistical expectation and variance, respectively |
| $\mathcal{CN}(0, \sigma^2)$ | Circularly symmetric complex Gaussian distribution |
| $\mathbb{C}^{M \times N}$ | A $M \times N$ complex-domain matrix |
| $\sim$ | "Distributed as" |

$M = 1$) and ($T \geq 2$, $M \geq 2$), the model reduces to SISO, SIMO, MISO and MIMO channel, respectively.

*Channel input-output relationship*: At time instant $i$ ($i = 1, 2, \ldots, n$), the signal received by the receiver is given by

$$\mathbf{Y}_i = \mathbf{h}\mathbf{X}_i + \boldsymbol{\eta}_i, \tag{1}$$

where the elements of $\boldsymbol{\eta}_i \in \mathbb{C}^{M \times 1}$ are independent identically distributed (i.i.d.) as $\mathcal{CN}(0, \sigma^2)$, $\mathbf{h} \in \mathbb{C}^{M \times T}$ is the channel gain of transmitter-receiver channel.

For convenience, in the remainder of this paper, channel gain $\mathbf{h}$ of the SISO/SIMO/MISO/MIMO channel is respectively denoted by $h_{\mathrm{siso}} \in \mathbb{C}^{1 \times 1}$, $\mathbf{h}_{\mathrm{simo}} \in \mathbb{C}^{M \times 1}$ ($M \geq 2$), $\mathbf{h}_{\mathrm{miso}} \in \mathbb{C}^{1 \times T}$ ($T \geq 2$) and $\mathbf{h}_{\mathrm{mimo}} \in \mathbb{C}^{M \times T}$ ($T \geq 2$, $M \geq 2$).

**Definition 1**: A $(n, \lvert \mathcal{W} \rvert, P)$-code with average power constraint consists of:

- The transmitted message $W$ is uniformly distributed over a finite set $\mathcal{W} = \{1, 2, \ldots, \lvert \mathcal{W} \rvert\}$.

- For the encoder: the output signal $\mathbf{X}_i = f(W, \mathbf{h}, \mathbf{Y}^{i-1})$ meets the average power constraint

$$\frac{1}{n} \sum_{i=1}^{n} E[\mathbf{X}_i^{\mathcal{H}} \mathbf{X}_i] \leq P, \tag{2}$$

where $f(\cdot)$ is the transmitter's deterministic encoding function.

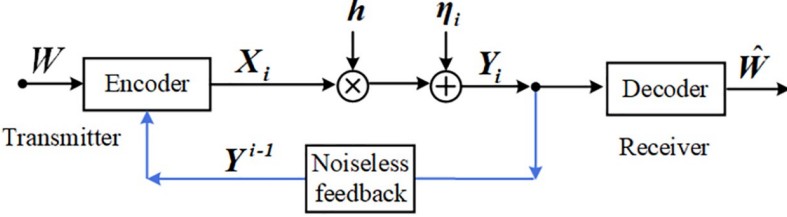

**Fig 1. The SISO/SIMO/MISO/MIMO systems ($T \geq 1$, $M \geq 1$) with noiseless feedback.**

- For the decoder: the output signal $\hat{w} = \varphi(\mathbf{Y}^n, \mathbf{h})$, here $\varphi$ is a decoding function for the receiver.

**Definition 2**: For the $(n, |\mathcal{W}|, P)$-code defined in the above Definition 1, the average decoding error probability is defined as

$$P_e = \frac{1}{|\mathcal{W}|} \sum_{W \in \mathcal{W}} \Pr\{\psi(\mathbf{Y}^n, \mathbf{h}) \neq W | W \text{ sent}\}. \tag{3}$$

The $(n, \varepsilon)$-transmission rate $R(n, \varepsilon)$ is achievable if for given blocklength $n$ and error probability $\varepsilon$, there is a $(n, |\mathcal{W}|, P)$-code introduced in Definition 1 such that

$$\frac{\log |\mathcal{W}|}{n} = R(n, \varepsilon), \quad P_e \leq \varepsilon, \tag{4}$$

and the maximal rate $R^*(n, \varepsilon)$ is the maximum rate defined in (4). In addition, the capacity is defined as

$$C = \lim_{N \to \infty} \lim_{\varepsilon \to 0} R^\star(n, \varepsilon). \tag{5}$$

## 3 SK-type FBL schemes for multi-antenna channels with noiseless feedback link

### 3.1 A SK-type FBL scheme for the SISO case ($T = M = 1$)

For the SISO channel, at time $i$ ($i = 1, 2, \ldots, n$), the signal received by the receiver is given by

$$Y_i = h_{\text{siso}} X_i + \eta_i, \tag{6}$$

where $h_{\text{siso}} \in \mathbb{C}^{1 \times 1}$ is the channel gain between the transmitter and the receiver channel, $X_i \in \mathbb{C}^{1 \times 1}$, and $\eta_i \in \mathbb{C}^{1 \times 1} \sim \mathcal{CN}(0, \sigma^2)$.

**Theorem 1**. For given $\varepsilon$ and $n$, an achievable rate $R_{\text{siso}}(n, \varepsilon)$ for the SISO channel with noiseless feedback link is given by

$$R_{\text{siso}}(n, \varepsilon) = C_{\text{siso}} - \underbrace{\frac{1}{n} \log \left[ \frac{1}{3} \left( Q^{-1} \left( \frac{\varepsilon}{4} \right) \right)^2 \left( \frac{\sigma^2}{P|h_{\text{siso}}|^2} + 1 \right) \right]}_{\text{Loss caused by given } (n, \varepsilon)}, \tag{7}$$

where $C_{\text{siso}} = \log \left( 1 + \frac{|h_{\text{siso}}|^2 P}{\sigma^2} \right)$ is the capacity of the SISO channel [14]. Since feedback does not increase the capacity of the memoryless channels, $C_{\text{siso}}$ is also the capacity of the SISO channel with noiseless feedback, which serves as an upper bound on $R_{\text{siso}}(n, \varepsilon)$.

**Proof**.

The signal received by the receiver in (6) can be expressed as

$$Y_{\text{R},i} + jY_{\text{I},i} = (h_{\text{R}} + jh_{\text{I}})(X_{\text{R},i} + jX_{\text{I},i}) + \eta_{\text{R},i} + j\eta_{\text{I},i}, \tag{8}$$

where $j$ is the imaginary unit, and the subscripts $R$ and $I$ respectively denote the real part and imaginary part of the original complex-domain elements.

According to Eq (8), we have

$$y_i' = X_{\text{R},i} + \eta_i', \quad y_i'' = X_{\text{I},i} + \eta_i'', \tag{9}$$

where

$$y'_i = \frac{h_{\mathrm{R}} Y_{\mathrm{R},i} + h_{\mathrm{I}} Y_{\mathrm{I},i}}{h_{\mathrm{R}}^2 + h_{\mathrm{I}}^2}, \quad y''_i = \frac{h_{\mathrm{R}} Y_{\mathrm{I},i} - h_{\mathrm{I}} Y_{\mathrm{R},i}}{h_{\mathrm{R}}^2 + h_{\mathrm{I}}^2},$$

$$\eta'_i = \frac{h_{\mathrm{R}} \eta_{\mathrm{R},i} + h_{\mathrm{I}} \eta_{\mathrm{I},i}}{h_{\mathrm{R}}^2 + h_{\mathrm{I}}^2}, \quad \eta''_i = \frac{h_{\mathrm{R}} \eta_{\mathrm{I},i} - h_{\mathrm{I}} \eta_{\mathrm{R},i}}{h_{\mathrm{R}}^2 + h_{\mathrm{I}}^2}. \tag{10}$$

Therefore, we conclude that the complex-domain signals and noise can be splitted into two real-domain signals and noise. Here note that the power constraints of $X_{\mathrm{R},i}$ and $X_{\mathrm{I},i}$ respectively meet $P_{\mathrm{R}}$ and $P_{\mathrm{I}}$, and $P_{\mathrm{R}} = P_{\mathrm{I}} = \frac{1}{2} P$. Moreover, from Eq (10), for the channel noise, we have $\mathrm{E}(\eta'_i) = 0$, $\mathrm{E}(\eta''_i) = 0$, $\mathrm{Var}(\eta'_i) = \frac{\sigma^2}{2|h_{\mathrm{siso}}|^2}$, and $\mathrm{Var}(\eta''_i) = \frac{\sigma^2}{2|h_{\mathrm{siso}}|^2}$.

The coding procedure of the two obtained sub-channels is analogous, hence we only give the detail coding procedure for one of the sub-channels in the following.

*Encoding procedure*:

For given $n$ and $\varepsilon$, let $|\mathcal{W}| = 2^{nR(n,\varepsilon)}$ and the message $W = (W_{\mathrm{R}}, W_{\mathrm{I}})$, here the values taken by $W_{\mathrm{R}}$ and $W_{\mathrm{I}}$ respectively satisfy $\mathcal{W}_{\mathrm{R}} = \{1, 2, ..., 2^{nR_{\mathrm{R}}}\}$ and $\mathcal{W}_{\mathrm{I}} = \{1, 2, ..., 2^{nR_{\mathrm{I}}}\}$, and the transmission rate satisfies

$$R_{\mathrm{R}} + R_{\mathrm{I}} = R(n, \varepsilon). \tag{11}$$

Splitted $[-0.5, 0.5]$ into $2^{nR_{\mathrm{R}}}$ equally spaced sub-intervals, note that the center of the each sub-interval is mapped to a value in $\mathcal{W}_{\mathrm{R}}$. Let $\beta_{\mathrm{R}}$ be the center of the obtained sub-interval w.r.t. the message $W_{\mathrm{R}}$.

At time instant 1, the transmitter sends the signal

$$X_{\mathrm{R},1} = \sqrt{12 P_{\mathrm{R}}} \beta_{\mathrm{R}}. \tag{12}$$

At the end of time instant 1, the receiver receives the signal $Y_1$, and then sends the signal $Y_1$ back to the transmitter by noiseless feedback link.

At time instant 2, the transmitter receives the feedback signal $Y_1$, obtains the signal $y'_1$ by using Eqs (8)–(10), and computes

$$\frac{y'_1}{\sqrt{12 P_{\mathrm{R}}}} = \beta_{\mathrm{R}} + \frac{\eta'_1}{\sqrt{12 P_{\mathrm{R}}}} = \beta_{\mathrm{R}} + \varepsilon_{\mathrm{R},1}, \tag{13}$$

where the receiver's estimation error in the last time instant is $\varepsilon_{\mathrm{R},1} = \frac{\eta'_1}{\sqrt{12 P_{\mathrm{R}}}}$. Then the transmitter sends the signal

$$X_{\mathrm{R},2} = \sqrt{\frac{P_{\mathrm{R}}}{\alpha_{\mathrm{R},1}}} \varepsilon_{\mathrm{R},1}, \tag{14}$$

where the variance of the estimation error is $\alpha_{\mathrm{R},1} = \mathrm{Var}(\varepsilon_{\mathrm{R},1})$.

At time instant $i + 1$ ($i = 2, 3, \ldots, n$), the transmitter receives the feedback signal $Y_i$, obtains the signal $y'_i$ by using Eqs (8)–(10), and computes the receiver's estimation error in the last time instant

$$\varepsilon_{\mathrm{R},i} = \varepsilon_{\mathrm{R},i-1} - \frac{E(y'_i \varepsilon_{\mathrm{R},i-1})}{E(y'_i)^2} y'_i. \tag{15}$$

Then the transmitter sends the signal,

$$X_{R,i+1} = \sqrt{\frac{P_R}{\alpha_{R,i}}} \varepsilon_{R,i}, \tag{16}$$

where $\alpha_{R,i} = \text{Var}(\varepsilon_{R,i})$.

The general term for $\alpha_{R,i}$ is given in the following Lemma 1, and note that the term is used in the procedure of the decoding error probability analysis.

**Lemma 1**.

For $\alpha_{R,i}$ ($i = 2, 3, \ldots, n$), the general term satisfies

$$\alpha_{R,i} = \frac{\sigma^2}{12P|h_{siso}|^2} \left\{ \frac{\sigma^2}{P|h_{siso}|^2 + \sigma^2} \right\}^{i-1}. \tag{17}$$

**proof**. The proof of Lemma 1 is similar to that of [13], hence the detailed proof is omitted here.

*Decoding procedure*:

At time instant 1, the receiver receives the signal $Y_1$, obtains the equivalent signal $y_1'$ by using Eqs (8)–(10), and the first estimation of $\beta_R$ is

$$\hat{\beta}_{R,1} = \frac{y_1'}{\sqrt{12P_R}} = \beta_R + \frac{\eta_1'}{\sqrt{12P_R}} = \beta_R + \varepsilon_{R,1}. \tag{18}$$

At time instant $i$ ($i = 2, 3, \ldots, n$), after receiving the signal $Y_i$, the receiver obtains the equivalent signal $y_i'$ by using (8)–(10), and the $i$-th updated estimation is given by

$$\hat{\beta}_{R,i} = \hat{\beta}_{R,i-1} - \frac{E(y_i' \varepsilon_{R,i-1})}{E(y_i')^2} y_i'. \tag{19}$$

From Eqs (15), (18) and (19), we obtain that for time instant $i = n$, the final estimation satisfies

$$\hat{\beta}_{R,n} = \varepsilon_{R,n} + \beta_R. \tag{20}$$

*Decoding error probability analysis*:

The decoding error probability $P_e$ of $W$ is bounded as follows, let $P_e \leq P_{e,R} + P_{e,I}$, here $P_{e,R}$ and $P_{e,I}$ are respectively the decoding error probabilities w.r.t. the message $W_R$ and $W_I$. From Eq (20) and the definition of the mapping value, we have

$$P_{e,R} \leq Pr\left\{ |\varepsilon_{R,n}| \geq \frac{1}{2 \times 2^{nR_R}} \right\} \overset{(a)}{\leq} 2Q\left( \frac{1}{2 \times 2^{nR_R} \sqrt{\alpha_{R,n}}} \right)$$

$$\overset{(b)}{=} 2Q\left( \frac{2^{-n\left[R_R - \frac{1}{2}\log\left(1 + \frac{|h_{siso}|^2 P}{\sigma^2}\right)\right]}}{2\sqrt{\frac{\sigma^2}{12P|h_{siso}|^2}\left(1 + \frac{|h_{siso}|^2 P}{\sigma^2}\right)}} \right), \tag{21}$$

where (a) is due to the fact that $Q(x)$ is the Gaussian $Q$-function, and (b) is based on the above Lemma 1. From Eq (21), for given $\varepsilon$ and $n$, we derive that if the rate satisfies

$$R_R \leq \frac{\log\left(1 + \frac{|h_{siso}|^2 P}{\sigma^2}\right)}{2} - \frac{\log\left\{2Q^{-1}\left(\frac{\varepsilon}{4}\right)\sqrt{\frac{\sigma^2}{12P|h_{siso}|^2} + \frac{1}{12}}\right\}}{n}, \tag{22}$$

$P_{e,R} \leq \varepsilon$ is guaranteed. Analogously, we derive that

$$R_I \leq \frac{\log\left(1 + \frac{|h_{siso}|^2 P}{\sigma^2}\right)}{2} - \frac{\log\left\{2Q^{-1}\left(\frac{\varepsilon}{4}\right)\sqrt{\frac{\sigma^2}{12P|h_{siso}|^2} + \frac{1}{12}}\right\}}{n}. \tag{23}$$

From Eqs (22), (23) and (11), we obtain $R(n, \varepsilon) = R_R + R_I$, and define $R(n, \varepsilon) = R_{siso}(n, \varepsilon)$, we derive the transmission rate $R_{siso}(n, \varepsilon)$ given in (7).

**Remark 1**. When the blocklength $n$ tends to infinity, the transmission rate $R(n, \varepsilon)$ of the SISO channel (see Eq (7)) approaches

$$\lim_{n \to \infty} R_{siso}(n, \varepsilon) = \log\left(1 + \frac{|h_{siso}|^2 P}{\sigma^2}\right) = C_{siso}. \tag{24}$$

## 3.2 A SK-type FBL scheme for the SIMO case ($T = 1, M \geq 2$)

For the SIMO case, at time $i$ ($i = 1, 2, \ldots, n$), the signal received by the receiver is given by

$$\mathbf{Y}_i = \mathbf{h}_{simo} X_i + \boldsymbol{\eta}_i, \tag{25}$$

where $\mathbf{h}_{simo} \in \mathbb{C}^{M \times 1}$ is channel of the SIMO case, $X_i \in \mathbb{C}^{1 \times 1}$, and distribution for the elements of $\boldsymbol{\eta}_i \in \mathbb{C}^{M \times 1}$ are i.i.d. as $\mathcal{CN}(0, \sigma^2)$.

**Theorem 2**. For given $\varepsilon$ and $n$, an achievable rate $R_{simo}(n, \varepsilon)$ for the SIMO channel with noiseless feedback link is given by

$$R_{simo}(n, \varepsilon) = C_{simo} - \underbrace{\frac{1}{n} \log\left[\left(Q^{-1}\left(\frac{\varepsilon}{4}\right)\right)^2 \left(\frac{\sigma^2}{3P\|\mathbf{h}_{simo}\|^2} + \frac{1}{3}\right)\right]}_{\text{Loss caused by given } (n, \varepsilon)}, \tag{26}$$

where $C_{simo} = \log\left(1 + \frac{\|\mathbf{h}_{simo}\|^2 P}{\sigma^2}\right)$ is the capacity of the SIMO channel [14]. Since feedback does not increase the capacity of the memoryless channels, $C_{simo}$ is also the capacity of the SIMO channel with noiseless feedback, which serves as an upper bound on $R_{simo}(n, \varepsilon)$.

**proof**.

The signal in Eq (25) can be rewritten as

$$\mathbf{h}_{simo}^{\mathcal{H}} \mathbf{Y}_i = \mathbf{h}_{simo}^{\mathcal{H}} \mathbf{h}_{simo} X_i + \mathbf{h}_{simo}^{\mathcal{H}} \eta_i = \|\mathbf{h}_{simo}\|^2 X_i + \mathbf{h}_{simo}^{\mathcal{H}} \eta_i, \tag{27}$$

where $\mathbf{h}_{simo} \in \mathbb{C}^{M \times 1}, \eta_i \in \mathbb{C}^{M \times 1}, X_i \in \mathbb{C}^{1 \times 1}, \mathbf{h}_{simo}^{\mathcal{H}} \eta_i \in \mathbb{C}^{1 \times 1} \sim \mathcal{CN}(0, \|\mathbf{h}_{simo}\|^2 \sigma^2), \mathbf{h}_{simo}^{\mathcal{H}} \mathbf{Y}_i \in \mathbb{C}^{1 \times 1}$.

Replacing $h_{siso}$ by $\|\mathbf{h}_{simo}\|^2$, $\eta_i$ by $\mathbf{h}_{simo}^{\mathcal{H}} \eta_i$, $Y_i$ by $\mathbf{h}_{simo}^{\mathcal{H}} \mathbf{Y}_i$, the SIMO channel can be transformed into the SISO channel defined in Eq (6). Hence along the lines of the coding procedure in the above subsection, it is not difficult to show that the transmission rate $R_{simo}(n, \varepsilon)$ given in (26) is achievable. Therefore the proof of Theorem 2 is completed.

**Remark 2**. When $n$ tends to infinity, the transmission rate $R_{simo}(n, \varepsilon)$ of the SIMO channel (see Eq (26)) approaches

$$\lim_{n \to \infty} R_{simo}(n, \varepsilon) = \log\left(1 + \frac{\|\mathbf{h}_{simo}\|^2 P}{\sigma^2}\right) = C_{simo}. \tag{28}$$

### 3.3 A SK-type FBL scheme for the MISO case ($T \geq 2$, $M = 1$)

For the MISO channel, at time instant $i$ ($i = 1, 2, \ldots, n$), the signal received by the receiver is given by

$$Y_i = \mathbf{h}_{\mathrm{miso}}\mathbf{X}_i + \eta_i, \tag{29}$$

where $\mathbf{h}_{\mathrm{miso}} \in \mathbb{C}^{1 \times T}$, $X_i \in \mathbb{C}^{T \times 1}$, and $\eta_i \in \mathbb{C}^{1 \times 1} \sim \mathcal{CN}(0, \sigma^2)$.

**Theorem 3**. For given $\varepsilon$ and $n$, an achievable rate $R_{\mathrm{miso}}(n, \varepsilon)$ for the MISO channel with noiseless feedback is given by

$$R_{\mathrm{miso}}(n, \varepsilon) = C_{\mathrm{miso}} - \underbrace{\frac{1}{n} \log \left[ \left( Q^{-1}\left(\frac{\varepsilon}{4}\right) \right)^2 \left( \frac{\sigma^2}{3P\|\mathbf{h}_{\mathrm{miso}}\|^2} + \frac{1}{3} \right) \right]}_{\textbf{Loss caused by given } (n, \varepsilon)}, \tag{30}$$

where $C_{\mathrm{miso}} = \log\left(1 + \frac{\|\mathbf{h}_{\mathrm{miso}}\|^2 P}{\sigma^2}\right)$ is the capacity of the MISO channel [14]. Since feedback does not increase the capacity of the memoryless channels, $C_{\mathrm{miso}}$ is also the capacity of the MISO channel with noiseless feedback, which serves as an upper bound on $R_{\mathrm{miso}}(n, \varepsilon)$.

**proof**.

Letting

$$\mathbf{X}_i = \frac{\mathbf{h}_{\mathrm{miso}}^{\mathcal{H}}}{\|\mathbf{h}_{\mathrm{miso}}\|} \tilde{X}_i \quad \Rightarrow \quad \tilde{X}_i = \frac{\mathbf{h}_{\mathrm{miso}}\mathbf{X}_i}{\|\mathbf{h}_{\mathrm{miso}}\|}, \tag{31}$$

where $\tilde{X}_i \in \mathbb{C}^{1 \times 1}$, $E(\mathbf{X}_i^{\mathcal{H}}\mathbf{X}_i) = E\left(\tilde{X}_i^{\mathcal{H}} \frac{\mathbf{h}_{\mathrm{miso}}}{\|\mathbf{h}_{\mathrm{miso}}\|} \frac{\mathbf{h}_{\mathrm{miso}}^{\mathcal{H}}}{\|\mathbf{h}_{\mathrm{miso}}\|} \tilde{X}_i\right) = E(\tilde{X}_i^{\mathcal{H}}\tilde{X}_i) = P$. The signal received by the receiver in Eq (29) is further expressed as

$$Y_i = \mathbf{h}_{\mathrm{miso}} \frac{\mathbf{h}_{\mathrm{miso}}^{\mathcal{H}}}{\|\mathbf{h}_{\mathrm{miso}}\|} \tilde{X}_i + \eta_i = \|\mathbf{h}_{\mathrm{miso}}\| \tilde{X}_i + \eta_i. \tag{32}$$

Replacing $h_{\mathrm{siso}}$ by $\|\mathbf{h}_{\mathrm{miso}}\|$, $X_i$ by $\tilde{X}_i$, we conclude that the MISO channel can be equivalent to the SISO channel defined in Eq (6). Then along the lines of the encoding-decoding procedure in the above subsection, it's not difficult to show that the transmission rate $R_{\mathrm{miso}}(n, \varepsilon)$ given in Eq (30) is achievable. Therefore the proof of Theorem 3 is completed.

**Remark 3**. When $n$ tends to infinity, the transmission rate $R_{\mathrm{miso}}(n, \varepsilon)$ of the MISO channel (see (30)) approaches

$$\lim_{n \to \infty} R_{\mathrm{miso}}(n, \varepsilon) = \log\left(1 + \frac{\|\mathbf{h}_{\mathrm{miso}}\|^2 P}{\sigma^2}\right) = C_{\mathrm{miso}}. \tag{33}$$

### 3.4 A SK-type FBL scheme for the MIMO case ($T \geq 2$, $M \geq 2$)

For the MIMO channel, at time instant $i$ ($i = 1, 2, \ldots, n$), the signal received by the receiver is given by

$$\mathbf{Y}_i = \mathbf{h}_{\mathrm{mimo}}\mathbf{X}_i + \eta_i, \tag{34}$$

where $\mathbf{h}_{\mathrm{mimo}} \in \mathbb{C}^{M \times T}$, $\mathbf{X}_i \in \mathbb{C}^{T \times 1}$, and the distribution for the elements of $\boldsymbol{\eta}_i \in \mathbb{C}^{M \times 1}$ are i.i.d. as $\mathcal{CN}(0, \sigma^2)$.

Based on the SVD (singular value decomposition) method, the matrix $\mathbf{h}_{\text{mimo}}$ can be expressed as

$$\mathbf{h}_{\text{mimo}} = \mathbf{U}\mathbf{D}\mathbf{V}^{\mathcal{H}}, \tag{35}$$

where $\mathbf{U} \in \mathbb{C}^{M \times M}$ and $\mathbf{V}^{\mathcal{H}} \in \mathbb{C}^{T \times T}$, here $\mathbf{U}\mathbf{U}^{\mathcal{H}} = \mathbf{I}_M$ and $\mathbf{V}\mathbf{V}^{\mathcal{H}} = \mathbf{I}_T$, $\mathbf{D} \in \mathbb{C}^{M \times T}$ is a diagonal matrix and $d_1, d_2 \ldots d_K$ are the diagonal elements, here $d_k$ ($k = 1, 2, \ldots, K$) is a real number [14, chapter 7], and $K$ is the minimum between $T$ and $M$.

**Theorem 4**. For given $\varepsilon$ and $n$, an achievable rate $R_{\text{mimo}}(n, \varepsilon)$ for the MIMO channel with noiseless feedback is given by

$$R_{\text{mimo}}(n, \varepsilon) = C_{\text{mimo}} - \underbrace{\min_{\sum_{k=1}^{K} P_k = P} \frac{1}{n} \sum_{k=1}^{K} \log\left[\left(\frac{\sigma^2}{3P_k d_k^2} + \frac{1}{3}\right)\left(Q^{-1}\left(\frac{\varepsilon}{4}\right)\right)^2\right]}_{\text{Loss caused by given }(n,\varepsilon)}, \tag{36}$$

where $C_{\text{mimo}} = \max_{\sum_{k=1}^{K} P_k = P} \sum_{k=1}^{K}\left[\log\left(1 + \frac{d_k^2 P_k}{\sigma^2}\right)\right]$ is the capacity of the MIMO channel [14]. Since feedback does not increase the capacity of the memoryless channels, $C_{\text{mimo}}$ is also the capacity of the MIMO channel with noiseless feedback, which serves as an upper bound on $R_{\text{mimo}}(n, \varepsilon)$.

**proof**.

From Eqs (34) and (35), the signal received by the receiver is rewritten by

$$\mathbf{Y}_i = \mathbf{U}\mathbf{D}\mathbf{V}^{\mathcal{H}}\mathbf{X}_i + \eta_i \Rightarrow \mathbf{U}^{\mathcal{H}}\mathbf{Y}_i = \mathbf{D}\mathbf{V}^{\mathcal{H}}\mathbf{X}_i + \mathbf{U}^{\mathcal{H}}\eta_i$$
$$\Rightarrow \mathbf{Y}'_i = \mathbf{D}\mathbf{X}'_i + \eta'_i, \tag{37}$$

where

$$\mathbf{Y}'_i = \mathbf{U}^{\mathcal{H}}\mathbf{Y}_i \in \mathbb{C}^{M \times 1}, \mathbf{X}'_i = \mathbf{V}^{\mathcal{H}}\mathbf{X}_i \in \mathbb{C}^{T \times 1}, \quad \eta'_i = \mathbf{U}^{\mathcal{H}}\eta_i \in \mathbb{C}^{M \times 1}, \tag{38}$$

$E[\mathbf{X}'^{\mathcal{H}}_i \mathbf{X}'_i] = E[\mathbf{X}^{\mathcal{H}}_i \mathbf{V}\mathbf{V}^{\mathcal{H}}\mathbf{X}_i] = E[\mathbf{X}^{\mathcal{H}}_i \mathbf{X}_i]$ $E[\eta'^{\mathcal{H}}_i \eta'_i] = E[\eta^{\mathcal{H}}_i \mathbf{U}\mathbf{U}^{\mathcal{H}}\eta_i] = E[\eta^{\mathcal{H}}_i \eta_i]$, and $\eta'_i \sim \mathcal{CN}(0, \sigma^2 \mathbf{I}_M)$. Since the matrix $\mathbf{D} \in \mathbb{C}^{M \times T}$ is a diagonal matrix and has diagonal elements $d_1, d_2 \ldots, d_K$, the MIMO channel (37) can be transformed into the following $K$ parallel sub-channels

$$Y'_{k,i} = d_k X'_{k,i} + \eta'_{k,i}, \quad i = 1, 2, \ldots, n, \quad k = 1, 2, \ldots, K. \tag{39}$$

The power allocated by the transmitter for each sub-channel meets $\sum_{k=1}^{K} P_k = P$. For given $n$ and $\varepsilon$, let $|\mathcal{W}| = 2^{nR(n,\varepsilon)}$, split $W = (W_1, W_2, \ldots, W_K)$, and further split $W_k = (W_{\text{R},k}, W_{\text{I},k})$ ($k = 1, 2, \ldots, K$), where $W_{\text{R},k}$ and $W_{\text{I},k}$ respectively take values in $\mathcal{W}_{\text{R},k} = \{1, 2, \ldots, 2^{nR_{\text{R},k}}\}$ and $\mathcal{W}_{\text{I},k} = \{1, 2, \ldots, 2^{nR_{\text{I},k}}\}$, where the transmission rate of the $k$-th sub-channel satisfies $R_k = R_{\text{R},k} + R_{\text{I},k}$, and the transmission rate $R(n, \varepsilon)$ for all $K$ sub-channels can be defined as

$$R(n, \varepsilon) = \sum_{k=1}^{K} R_k. \tag{40}$$

Along the lines of the coding procedure in the above subsection, the transmission rate $R_k$ is given by

$$R_k = R_{\text{R},k} + R_{\text{I},k} = \left\{\log\left(1 + \frac{d_k^2 P_k}{\sigma^2}\right) - \frac{1}{n}\log\left[\left(\frac{\sigma^2}{3P_k d_k^2} + \frac{1}{3}\right)\left(Q^{-1}\left(\frac{\varepsilon}{4}\right)\right)^2\right]\right\}. \tag{41}$$

Combining Eqs (40) and (41), and power allocating mentioned above, the transmission rate $R_{\mathrm{mimo}}(n, \varepsilon)$ given in (36) can be derived, hence the proof of Theorem 4 is completed.

**Remark 4**. When $n$ tends to infinity, the transmission rate $R_{\mathrm{mimo}}(n, \varepsilon)$ of the MIMO channel (see (36)) approaches

$$\lim_{n \to \infty} R_{\mathrm{mimo}}(n, \varepsilon) = \max_{\sum_{k=1}^{K} P_k = P} \sum_{k=1}^{K} \left\{ \log \left( 1 + \frac{d_k^2 P_k}{\sigma^2} \right) \right\} = C_{\mathrm{mimo}}. \tag{42}$$

The following Remark 5 explains the significance and importance of the results given in Theorems 1-4.

**Remark 5**. In the literature, the classical SK scheme has been shown to be a good finite blocklenth coding scheme for the AWGN channel with feedback since its decoding error probability doubly exponentially decays to zero as the coding blocklength tends to infinity. However, this well-performed scheme cannot be applied to practical wireless multi-antenna channels since it is only designed for the AWGN channel. In this paper, we extend the SK scheme to various wireless channels, and characterize the achievable transmission rates of these extended schemes (see Theorems 1-4), which provides a way for the application of the SK-type scheme to practical wireless communication systems. In addition, the established achievable rates in Theorems 1-4 can also be viewed as the fundamental limit of the achievable rates of the SK-type schemes for SISO/SIMO/MISO/MIMO channel with noisy feedback.

## 4 Simulation results

Let the elements of the channel gains **h** and **g** be i.i.d. as $\mathcal{CN}(0, 1)$. All results are calculated based on an average of 1000 independent channel realizations.

For SISO/SIMO/MISO/MIMO channels with noiseless feedback link, Figs 2–5 show that when the blocklengths are greater than 50, the gap between the transmission rates, respectively derived by infinite blocklength $n$ and finite blocklength $n$, is significantly small. This observation indicates that our proposed feedback-based coding schemes can approximate the SISO/SIMO/MISO/MIMO channel capacities when the blocklength is sufficiently short.

## 5 Conclusions and future work

In this paper, the constructive SK-type FBL feedback coding schemes for the wireless static channels are proposed. Simulation results show that the proposed feedback-based schemes almost approach the SISO/MISO/SIMO/MIMO channel capacities when the blocklength is sufficiently short. One possible future work is to extend the proposed schemes in this paper to the multi-user channels in the presence of eavesdropper.

## Author Contributions

**Conceptualization:** Tao Song.

**Supervision:** Tao Song.

**Writing – original draft:** Qiang Huang.

**Writing – review & editing:** Tao Song.

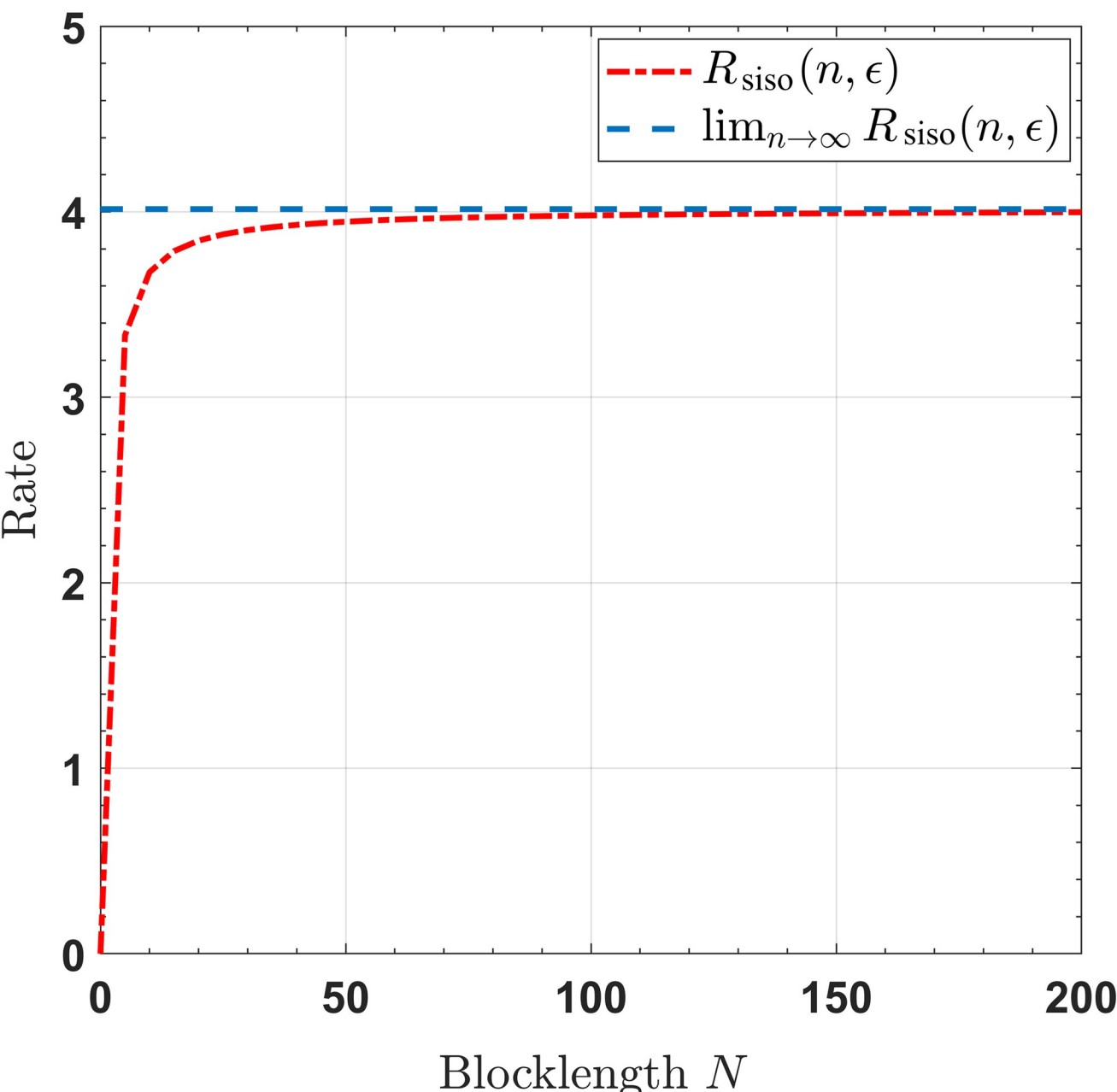

**Fig 2. The rates for the SISO channel, with $P = 10$, $\sigma^2 = 1$, $\epsilon = 10^{-6}$.**

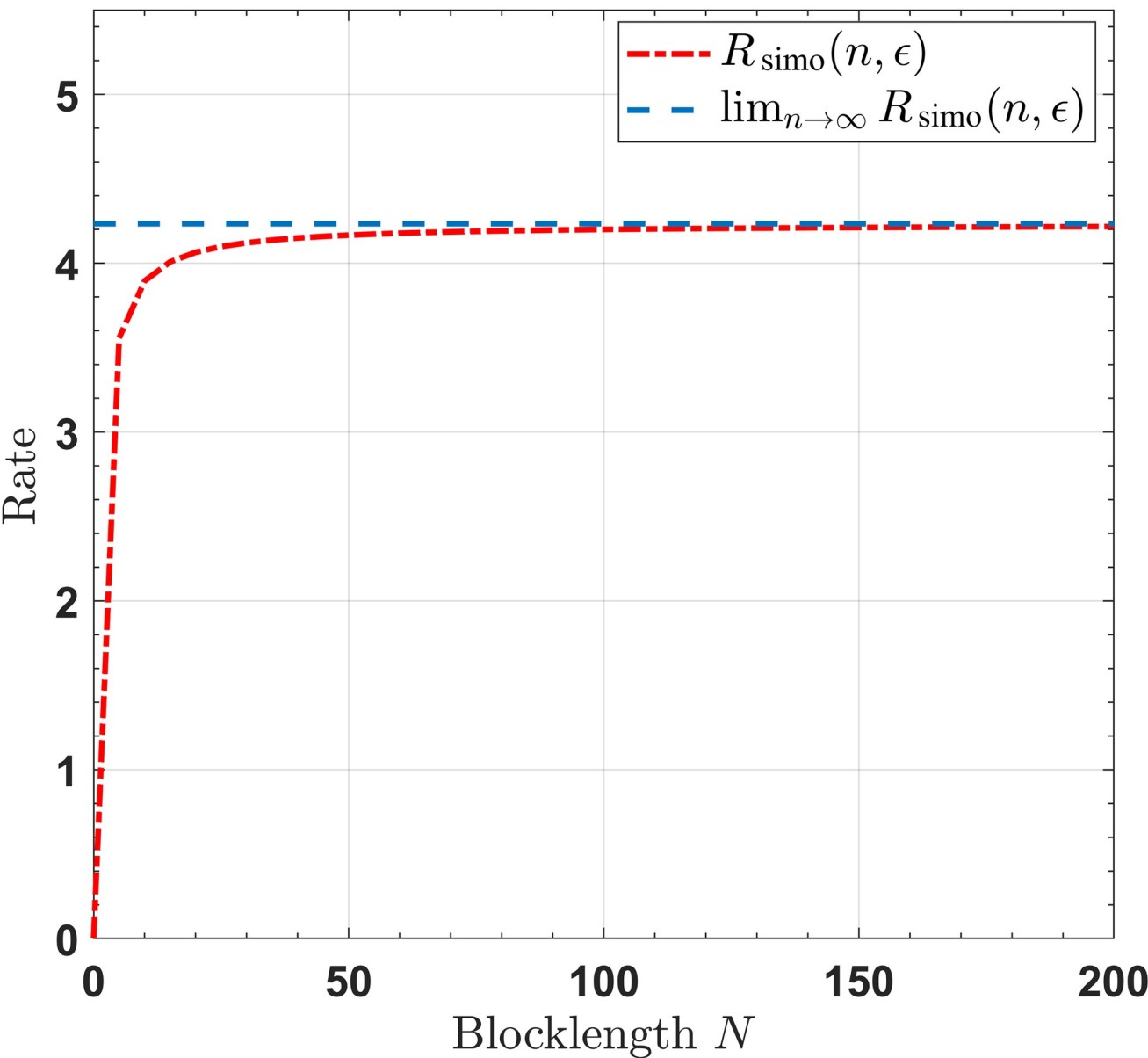

**Fig 3. The rates for the SIMO channel, with $M = 4$, $P = 10$, $\sigma^2 = 1$, $\epsilon = 10^{-6}$.**

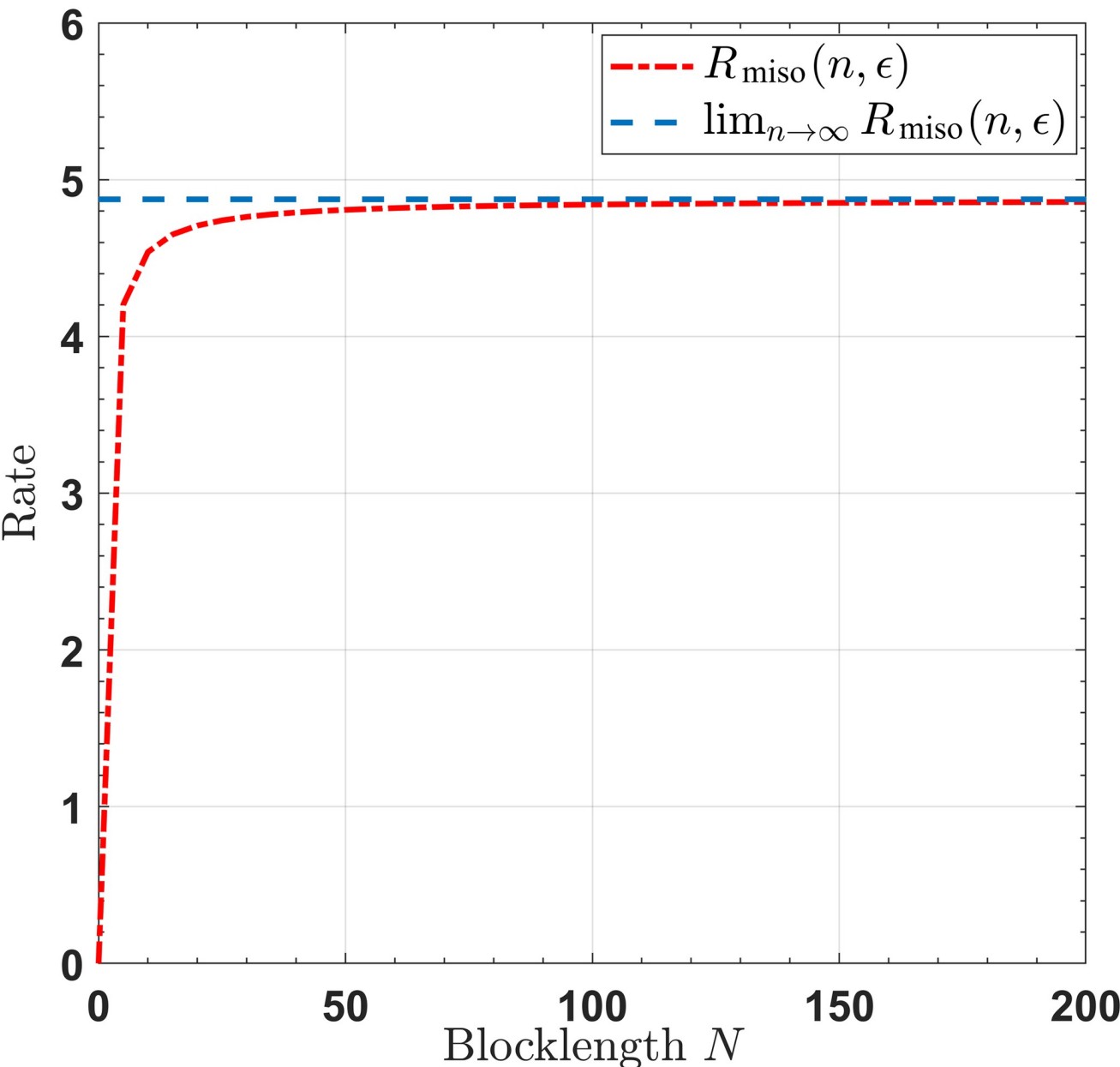

**Fig 4. The rates for the MISO channel, with $T = 4$, $P = 10$, $\sigma^2 = 1$, $\epsilon = 10^{-6}$.**

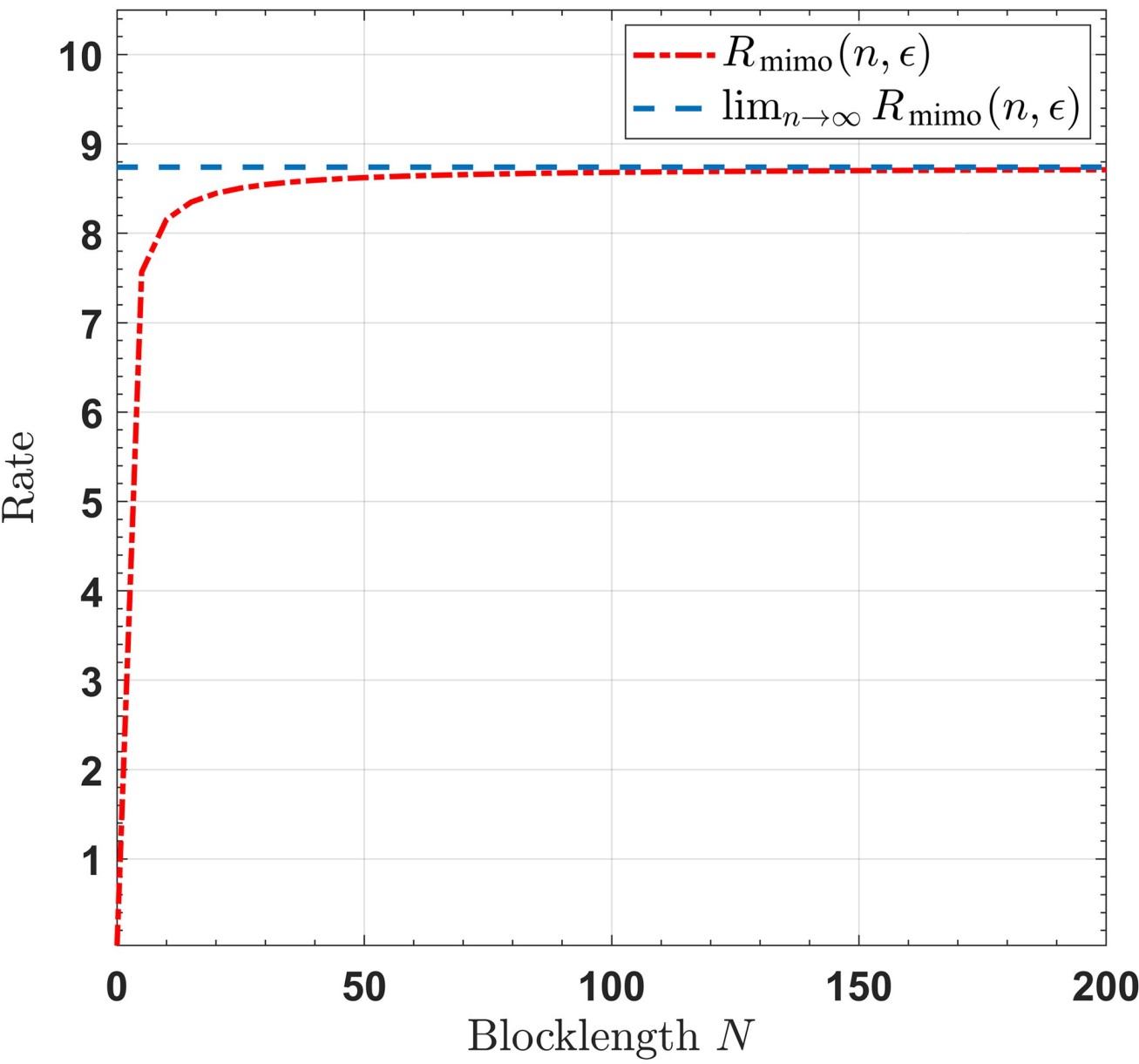

**Fig 5. The rates for the MIMO channel, with $T = 4$, $M = 4$, $P = 10$, $\sigma^2 = 1$, $\epsilon = 10^{-6}$.**

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
