## [Decision Letter · Decision Letter 0]

11 Oct 2023

PONE-D-23-29974Finite Blocklength Feedback Approach for Multi-Antenna Wireless ChannelsPLOS ONE

Dear Dr. Song,

Thank you for submitting your manuscript to PLOS ONE. After careful consideration, we feel that it has merit but does not fully meet PLOS ONE’s publication criteria as it currently stands. Therefore, we invite you to submit a revised version of the manuscript that addresses the points raised during the review process.

We look forward to receiving your revised manuscript.

Kind regards,

Praveen Kumar Donta, Ph.D.

Academic Editor

PLOS ONE

Journal Requirements:

Reviewers' comments:

Reviewer's Responses to Questions

**Comments to the Author**

1. Is the manuscript technically sound, and do the data support the conclusions?

Reviewer #1: No

Reviewer #2: Yes

Reviewer #3: No

2. Has the statistical analysis been performed appropriately and rigorously? 

Reviewer #1: No

Reviewer #2: Yes

Reviewer #3: No

3. Have the authors made all data underlying the findings in their manuscript fully available?

Reviewer #1: No

Reviewer #2: Yes

Reviewer #3: No

4. Is the manuscript presented in an intelligible fashion and written in standard English?

Reviewer #1: Yes

Reviewer #2: Yes

Reviewer #3: Yes

5. Review Comments to the Author

Reviewer #1: Major comments

1, When introducing the FBL analyses, the authors ignored the work of Strassen [1] and down- graded the work of Hayashi [2, 3]. In fact, second-order asymptotics was first studied by Strassen and first revived by Hayashi using information spectrum methods [4]. The notion of “second-order coding rate” was also coined by Hayashi. Furthermore, the equation in the FBL analysis is not numbered and the third-order term has been characterized for certain channels. The authors could refer to [5, 6] for details on FBL analyses and history.

2,The definition of the fundamental limit in (4) is incomplete. It only specifies the achievability part. However, a formal definition should specify the optimal rate, as in [7, 8].

3, What is the significance of Theorem 1 and why is it important? Compared with [8], the results appear in very different forms. What is the intuition here and why is it reasonable. Furthermore, since Theorem 1 is only an achievability result, what is the converse bound? How large is the gap? All these questions are critical for finite blocklength analyses. Finally, the proof of Theorem 1 appears rather strange in the perspective of finite blocklength analyses. The above comments for Theorem 1 apply to all other theorems in the paper.

4, The novelty of the paper is unclear to me because the analysis is not proved tight or connected with any prior art in the same direction. The analyses appear strange to me.

Minor comments

1, The abbreviations SISO/SIMO/MISO/MIMO appear all undefined.

2, The additional space after the dash between two references are unnecessary.

3, In code definition, the parameter comes first and the phrase “code” appears after (cf. [7]).

4, The non-math contents in math equations including “Pr” and “SISO” should be in the mathrm command so that they are italic in math environment.

5, The bracket in Eq. (6) (and other equations) is not large enough.

References

[1] V. Strassen, “Asymptotische abschätzungen in shannons informationstheorie,” in Trans. Third Prague Conf. Information Theory, 1962, pp. 689–723.

[2] M. Hayashi, “Information spectrum approach to second-order coding rate in channel coding,” IEEE Trans. Inf. Theory, vol. 55, no. 11, pp. 4947–4966, 2009.

1

[3] ——, “Second-order asymptotics in fixed-length source coding and intrinsic randomness,” IEEE Trans. Inf. Theory, vol. 54, no. 10, pp. 4619–4637, 2008.

[4] T. S. Han, Information-Spectrum Methods in Information Theory. Springer Berlin Heidelberg, 2003.

[5] V. Y. F. Tan, “Asymptotic estimates in information theory with non-vanishing error probabili- ties,” Foundations and Trends ® in Communications and Information Theory, vol. 11, no. 1–2, pp. 1–184, 2014.

[6] L. Zhou and M. Motani, “Finite blocklength lossy source coding for discrete memoryless sources,” Foundations and Trends ® in Communications and Information Theory, vol. 20, no. 3, pp. 157–389, 2023.

[7] Y. Polyanskiy, H. V. Poor, and S. Verdú, “Channel coding rate in the finite blocklength regime,” IEEE Trans. Inf. Theory, vol. 56, no. 5, pp. 2307–2359, 2010.

[8] Y. Polyanskiy, H. V. Poor, and S. Verdú, “Feedback in the non-asymptotic regime,” IEEE Trans. Inf. Theory, vol. 57, no. 8, pp. 4903–4925, 2011.

Reviewer #2: The authors have introduced constructive SK-type FBL feedback coding schemes for wireless fading channels.

Simulation results show that the proposed feedback-based schemes almost approach the SISO/MISO/SIMO/MIMO channel capacities with short blocklength. I consider that the manuscript shows relevant and well described findings in terms of the FBL coding scheme for the wireless multi-antenna fading channels.

Overall, it is well-written and comprehensible. I have a few minor suggestions based on my review:

1) Can the results of this paper (Theorem 1-Theorem 4) be expressed in terms of Shannon capacity, as in Polyanskiy et al.?

2) Some errors, e.g., in Figure 1, the feedback signal ``Yi-1” should be ``Yi-1”,

in line 133 of page 13, ``channels” should be ``channel”,

in Table I, the symbol “|.|” denotes modulus if applied to a complex number or cardinality if applied to a set, not only the cardinality of a set as the authors claim.

Reviewer #3: Major:

- Authors have derived a converse bound for a special type of coding scheme in the finite blocklength regime. However, the results are claimed to show the exact channel capacity, which is not true. Therefore, I advise the authors to re-write their claims.

- On the other hand, several other converse bounds for finite blocklength regime coding with feedback have already been published in the literature. Thus, comparison with them is also important to understand if the proposed bound is loose or tight compared to the others.

- Additionally, the inequality (a) in Eq. 20 is true for the asymptotic regime. Therefore, it is unclear if this inequality holds for finite blocklength. I advise the authors to revisit this issue and support their claims with additional references.

- When perfect channel state is available at the transmitter and receiver, it is possible to match the rate with the instantaneous channel capacity. In this case, the maximum achievable capacity is the ergodic capacity, not log(1+SNR). However, Eq. (23) contradicts with this and claims that the achievable capacity is the instantaneous capacity. Are the Authors sure in their remark? If yes, please support your claims. Similar contradictions also appear in the other Remarks for SIMO, MISO, MIMO.

- I do not understand what is plotted in the Simulation Results. For instance, in Fig.2 there are two lines. One is the classical Shannon capacity log(1+SNR). What about the other? Is this the output of Eq. 6? If yes, it is also claimed that results are calculated based on an average of 1000 independent channel realizations. Therefore, it cannot be Eq. 6 right? If it is the average of achievable rates of the proposed codes, I believe it is more important to show the comparison between instantaneous channel vs the achieved rate with the proposed codes in time for several different blocklength?

Minor:

- Wrong symbol in Eq. 3.

6. PLOS authors have the option to publish the peer review history of their article (what does this mean?). If published, this will include your full peer review and any attached files.

Reviewer #1: No

Reviewer #2: No

Reviewer #3: No

---

## [Author Response · Author response to Decision Letter 0]

9 Nov 2023

The response to the reviewers is in the attachment.

---

## [Decision Letter · Decision Letter 1]

23 Nov 2023

PONE-D-23-29974R1Finite Blocklength Feedback Approach for Multi-Antenna Wireless ChannelsPLOS ONE

Dear Dr. Song,

Thank you for submitting your manuscript to PLOS ONE. After careful consideration, we feel that it has merit but does not fully meet PLOS ONE’s publication criteria as it currently stands. Therefore, we invite you to submit a revised version of the manuscript that addresses the points raised during the review process.

We look forward to receiving your revised manuscript.

Kind regards,

Praveen Kumar Donta, Ph.D.

Academic Editor

PLOS ONE

Reviewers' comments:

Reviewer's Responses to Questions

**Comments to the Author**

1. If the authors have adequately addressed your comments raised in a previous round of review and you feel that this manuscript is now acceptable for publication, you may indicate that here to bypass the “Comments to the Author” section, enter your conflict of interest statement in the “Confidential to Editor” section, and submit your "Accept" recommendation.

Reviewer #1: (No Response)

Reviewer #2: All comments have been addressed

2. Is the manuscript technically sound, and do the data support the conclusions?

Reviewer #1: Partly

Reviewer #2: Yes

3. Has the statistical analysis been performed appropriately and rigorously? 

Reviewer #1: Yes

Reviewer #2: Yes

4. Have the authors made all data underlying the findings in their manuscript fully available?

Reviewer #1: Yes

Reviewer #2: Yes

5. Is the manuscript presented in an intelligible fashion and written in standard English?

Reviewer #1: Yes

Reviewer #2: Yes

6. Review Comments to the Author

Reviewer #1: The authors have addressed many of my comments. However, the reply to my comment 3 in the last round regarding the significance and importance of the results should be incorporated into the paper as remarks. Furthermore, the subscript such as siso in h_{siso} should be written as h_{\\mathrm{siso}}. There are many such errors to be corrected throughout the paper. Finally, in the updated introduction, it would be good to mention author names instead of a reference number when mentioning a result and there is an additional space of citations of two papers of Hayashi.

Reviewer #2: (No Response)

7. PLOS authors have the option to publish the peer review history of their article (what does this mean?). If published, this will include your full peer review and any attached files.

Reviewer #1: No

Reviewer #2: No

---

## [Decision Letter · Decision Letter 2]

18 Dec 2023

PONE-D-23-29974R2Finite Blocklength Feedback Approach for Multi-Antenna Wireless ChannelsPLOS ONE

Dear Dr. Song,

Thank you for submitting your manuscript to PLOS ONE. After careful consideration, we feel that it has merit but does not fully meet PLOS ONE’s publication criteria as it currently stands. Therefore, we invite you to submit a revised version of the manuscript that addresses the points raised during the review process.

We look forward to receiving your revised manuscript.

Kind regards,

Praveen Kumar Donta, Ph.D.

Academic Editor

PLOS ONE

Journal Requirements:

Reviewers' comments:

Reviewer's Responses to Questions

**Comments to the Author**

1. If the authors have adequately addressed your comments raised in a previous round of review and you feel that this manuscript is now acceptable for publication, you may indicate that here to bypass the “Comments to the Author” section, enter your conflict of interest statement in the “Confidential to Editor” section, and submit your "Accept" recommendation.

Reviewer #1: (No Response)

Reviewer #3: (No Response)

2. Is the manuscript technically sound, and do the data support the conclusions?

Reviewer #1: Yes

Reviewer #3: Yes

3. Has the statistical analysis been performed appropriately and rigorously? 

Reviewer #1: Yes

Reviewer #3: Yes

4. Have the authors made all data underlying the findings in their manuscript fully available?

Reviewer #1: Yes

Reviewer #3: Yes

5. Is the manuscript presented in an intelligible fashion and written in standard English?

Reviewer #1: Yes

Reviewer #3: Yes

6. Review Comments to the Author

Reviewer #1: Two minor comments:

1, The non-math content in the figure legends should also be addressed. Pay attention to Fig. 2 to Fig. 5.

2, When citing papers, use et al. only if the number of authors is greater than three. Otherwise, list the family names of all authors.

3, The real and imaginary parts notations R and I are not math content as well. Thus, all subscripts involving these two should be addressed as well. The authors should check carefully for all such minor mistakes and correct them all.

Reviewer #3: I appreciate the revisions made to the manuscript. However, there still seems to be a discrepancy in the sections where the authors claim to be studying fading channels. The assumption that the channel remains constant throughout the entire communication and is known to both the transmitter and receiver, as indicated in the latest comments, seems to simplify the system model to a basic AWGN channel, which potentially limits the scope of the proposed method for fading channels. Therefore, I recommend that the authors consider a more cautious tone in presenting their contributions to fading channels. Hence, the authors should modify the abstract, contributions, and system model sections and avoid emphasizing that the study proposes a method for fading channels.

7. PLOS authors have the option to publish the peer review history of their article (what does this mean?). If published, this will include your full peer review and any attached files.

Reviewer #1: No

Reviewer #3: No

---

## [Author Response · Author response to Decision Letter 2]

24 Dec 2023

The response to reviewers is in the attachment

---

## [Decision Letter · Decision Letter 3]

2 Jan 2024

PONE-D-23-29974R3Finite Blocklength Feedback Approach for Multi-Antenna Wireless ChannelsPLOS ONE

Dear Dr. Song,

Thank you for submitting your manuscript to PLOS ONE. After careful consideration, we feel that it has merit but does not fully meet PLOS ONE’s publication criteria as it currently stands. Therefore, we invite you to submit a revised version of the manuscript that addresses the points raised during the review process.

We look forward to receiving your revised manuscript.

Kind regards,

Praveen Kumar Donta, Ph.D.

Academic Editor

PLOS ONE

Journal Requirements:

Reviewers' comments:

Reviewer's Responses to Questions

**Comments to the Author**

1. If the authors have adequately addressed your comments raised in a previous round of review and you feel that this manuscript is now acceptable for publication, you may indicate that here to bypass the “Comments to the Author” section, enter your conflict of interest statement in the “Confidential to Editor” section, and submit your "Accept" recommendation.

Reviewer #1: All comments have been addressed

Reviewer #3: (No Response)

2. Is the manuscript technically sound, and do the data support the conclusions?

Reviewer #1: Yes

Reviewer #3: Yes

3. Has the statistical analysis been performed appropriately and rigorously? 

Reviewer #1: Yes

Reviewer #3: Yes

4. Have the authors made all data underlying the findings in their manuscript fully available?

Reviewer #1: Yes

Reviewer #3: No

5. Is the manuscript presented in an intelligible fashion and written in standard English?

Reviewer #1: Yes

Reviewer #3: Yes

6. Review Comments to the Author

Reviewer #1: (No Response)

Reviewer #3: Thank you for addressing my comment and making revisions to the manuscript. However, I would like to point out that the term 'quasi-static fading channel' does not accurately represent the assumptions outlined in the manuscript. The assumption mentioned, where the channel remains constant throughout the entire communication and is known to both the transmitter and receiver, simplifies the system model to that of a basic Additive White Gaussian Noise (AWGN) channel.

In a true quasi-static fading channel, while the channel conditions may change, they do so at a much slower rate compared to the transmission rate. Specifically, the channel coefficient is assumed to be an independent and identically distributed random variable according to some distribution but remains constant over the codeword transmission and it is unknown or partially known to the transmitter and receiver, which is also the case in the referenced papers.

If my explanations remain unclear to the authors, I recommend reading Chapter 5.4 of David Tse's "Fundamentals of Wireless Communication" for further clarification.

Hence, the mentioned assumption in the manuscript appears to lead to a scenario closer to a static channel rather than a quasi-static fading channel. Therefore, to maintain accuracy in the manuscript, it would be more appropriate to characterize the channel conditions as 'static channel' rather than using the term 'quasi-static fading channel'. I recommend revising the manuscript accordingly.

7. PLOS authors have the option to publish the peer review history of their article (what does this mean?). If published, this will include your full peer review and any attached files.

Reviewer #1: No

Reviewer #3: No

---

## [Author Response · Author response to Decision Letter 3]

4 Jan 2024

The response to reviewer is in the attachment.

---

## [Decision Letter · Decision Letter 4]

9 Jan 2024

PONE-D-23-29974R4Finite Blocklength Feedback Approach for Multi-Antenna Wireless ChannelsPLOS ONE

Dear Dr. Song,

Thank you for submitting your manuscript to PLOS ONE. After careful consideration, we feel that it has merit but does not fully meet PLOS ONE’s publication criteria as it currently stands. Therefore, we invite you to submit a revised version of the manuscript that addresses the points raised during the review process.

We look forward to receiving your revised manuscript.

Kind regards,

Praveen Kumar Donta, Ph.D.

Academic Editor

PLOS ONE

Journal Requirements:

Additional Editor Comments:

Please address the reviewer comments carefully.

Reviewers' comments:

Reviewer's Responses to Questions

**Comments to the Author**

1. If the authors have adequately addressed your comments raised in a previous round of review and you feel that this manuscript is now acceptable for publication, you may indicate that here to bypass the “Comments to the Author” section, enter your conflict of interest statement in the “Confidential to Editor” section, and submit your "Accept" recommendation.

Reviewer #3: (No Response)

2. Is the manuscript technically sound, and do the data support the conclusions?

Reviewer #3: Yes

3. Has the statistical analysis been performed appropriately and rigorously? 

Reviewer #3: Yes

4. Have the authors made all data underlying the findings in their manuscript fully available?

Reviewer #3: No

5. Is the manuscript presented in an intelligible fashion and written in standard English?

Reviewer #3: Yes

6. Review Comments to the Author

Reviewer #3: Once again, I would like to highlight that the term 'static fading channel' is still misleading. In wireless communication, a 'fading channel' typically refers to a scenario where the channel conditions vary over time, and using the term 'static' creates confusion. In fact, there is no such thing as 'static fading channel' in wireless communication. On the other hand, the term 'static fading channel' is still but very rarely used in the literature, however, when used, it consistently refers to quasi-static channels. Therefore, if the intention is to convey that the channel conditions remain constant during the entire communication, which is the case in the current manuscript, the term 'static' alone without 'fading' is accurate.

7. PLOS authors have the option to publish the peer review history of their article (what does this mean?). If published, this will include your full peer review and any attached files.

Reviewer #3: No

---

## [Author Response · Author response to Decision Letter 4]

9 Jan 2024

The response to reviewers is in the attachment.

---

## [Editor Report · Decision Letter 5]

15 Jan 2024

Finite Blocklength Feedback Approach for Multi-Antenna Wireless Channels

PONE-D-23-29974R5

Dear Dr. Song,

We’re pleased to inform you that your manuscript has been judged scientifically suitable for publication and will be formally accepted for publication once it meets all outstanding technical requirements.

Kind regards,

Praveen Kumar Donta, Ph.D.

Academic Editor

PLOS ONE
---

## [Editor Report · Acceptance letter]

2 Feb 2024

PONE-D-23-29974R5 

PLOS ONE

Dear Dr. Song, 

I'm pleased to inform you that your manuscript has been deemed suitable for publication in PLOS ONE. Congratulations! Your manuscript is now being handed over to our production team.

Kind regards, 

on behalf of

Dr. Praveen Kumar Donta 

Academic Editor

PLOS ONE